# *What's the Magic Word?* A Control Theory of LLM Prompting

## Abstract

Prompt engineering is crucial for deploying LLMs but is poorly understood mathematically. We formalize LLM systems as a class of discrete stochastic dynamical systems to explore prompt engineering through the lens of control theory. We investigate the reachable set of output token sequences $\mathcal{R}_y(\mathbf{x}_0)$ for which there exists a control input sequence $\mathbf{u}$ for each $\mathbf{y} \in \mathcal{R}_y(\mathbf{x}_0)$ that steers the LLM to output $\mathbf{y}$ from initial state sequence $\mathbf{x}_0$. We offer analytic analysis on the limitations on the controllability of self-attention in terms of reachable set, where we prove an upper bound on the reachable set of outputs $\mathcal{R}_y(\mathbf{x}_0)$ as a function of the singular values of the parameter matrices. We present complementary empirical analysis on the controllability of a panel of LLMs, including Falcon-7b, Llama-7b, and Falcon-40b. Our results demonstrate a lower bound on the reachable set of outputs $\mathcal{R}_y(\mathbf{x}_0)$ w.r.t. initial state sequences $\mathbf{x}_0$ sampled from the Wikitext dataset. We find that the correct next Wikitext token following sequence $\mathbf{x}_0$ is reachable over 97% of the time with prompts of $k \leq 10$ tokens. We also establish that the top 75 most likely next tokens, as estimated by the LLM itself, are reachable at least 85% of the time with prompts of $k \leq 10$ tokens. Intriguingly, short prompt sequences can dramatically alter the likelihood of specific outputs, even making the least likely tokens become the most likely ones. This control-centric analysis of LLMs demonstrates the significant and poorly understood role of input sequences in steering output probabilities, offering a foundational perspective for enhancing language model system capabilities.

## 1 Introduction

LLMs pre-trained on unsupervised next token prediction objectives exhibit unprecedented dynamic reprogrammability achieved through "prompting", often referred to as zero-shot learning (Brown et al., 2020; Wei et al., 2022; Hagendorff, 2023; Noever & McKee, 2023; OpenAI, 2023; 2022). These capabilities appear to emerge as the model's size, training data, and training time are scaled. The dynamic reprogrammability of LLMs is akin to the adaptable computational capacities observed in biological systems. This feature finds applications across domains such as machine translation (Wang et al., 2023a), code generation (Rozière et al., 2023), and chatbots Bai et al. (2022). A rigorous understanding of the prompt's influence over LLM generation would be of great utility for understanding LLMs and building more robust and capable systems leveraging LLMs.

Strategies for controlling pre-trained LLM generation today fall into three broad categories (Zhang et al., 2022):

1. **Input Optimization (Prompting):** Adjusting the input tokens (e.g., rewording the prompt) to improve subsequent text generation.

2. **Model Optimization:** Adjusting the weights of the network (e.g., fine-tuning, RLHF) to improve model behavior during inference.

3. **Post-processing:** Adjusting or re-ranking generated text (e.g., surrogate ranking algorithm).

Of all these approaches, input optimization (i.e., prompting) is the least invasive and lowest-cost method – and the least understood. Prompt optimization is also deeply connected to the zero-shot

capabilities of LLMs – the mysterious emergent capabilities of LLMs such as problem-solving, knowledge retrieval, reasoning, and apparent general intelligence (Bubeck et al., 2023). With such a view, we seek to characterize the controllability of LLMs via prompting.

## 1.1 CONTRIBUTION

We formalize LLM systems in the mathematical framework of control theory in Section 3. Our analysis focuses on the reachable set of outputs $\mathcal{R}_y(\mathbf{x}_0)$ for an LLM system. The reachable set is a fundamental concept in control theory that underlies notions of controllability, stability, and observability (cf. Appendix A). The reachable output set $R_y(\mathbf{x}_0)$ is the set of output sequences $\mathbf{y}$ for which there exists a control input sequence $\mathbf{u}^*$ that steers the LLM from initial state $\mathbf{x}_0$ to output $\mathbf{y}$ (cf. Definitions 3, 11).

Our analytic results in Section 4 prove an upper bound on the contents of the reachable output set for a self-attention head as a function of the singular values of its parameter matrices. Since self-attention is the only component in a transformer block where significant information is exchanged between token representations, this bound provides a foothold for analysis of LLM controllability from the perspective of mechanistic interpretability (e.g., Bricken et al. (2023); Chefer et al. (2021); Conmy et al. (2023)). Moreover, this bound represents a necessary condition for an output to be in the reachable set.

Our empirical results apply state-of-the-art prompt optimization techniques (Section 5.1) to demonstrate a lower bound on the contents of the reachable output set for a panel of LLMs, including Llama-7b (Touvron et al., 2023), Falcon-7b, and Falcon-40b (Almazrouei et al., 2023). Specifically, we sample initial states $\mathbf{x}_0$ from the Wikitext dataset (Merity et al., 2016) and probe the reachable output tokens $y$ under length-constrained control input sequences $\mathbf{u} : |\mathbf{u}| \leq k$. The length constraint $k$ is highly relevant for *optimal control* of LLMs, as prompts with fewer tokens require fewer computation and memory resources. We find that the reachable output set contains the "correct" next Wikitext token following $\mathbf{x}_0$ over 97% of the time with prompts of $k \leq 10$ tokens. We expand our analysis of the contents of $R_y(\mathbf{x}_0)$ by sampling target output tokens $y$ based on the LLMs initial estimate of output likelihood $P_{LM}(y|\mathbf{x}_0)$. We find that the top 75 most likely output tokens $y$ are reachable at least 85% of the time with prompts of $k \leq 10$ tokens. Intriguingly, some tokens drawn from the set of *least* likely outputs are controllable to the most likely output with $k \leq 4$ control input tokens. Our results suggest that prior likelihood-based metrics, such as cross-entropy loss, cannot guarantee exclusion from the reachable set, emphasizing the gap in our current understanding of LLM systems and control. Implications of our results and open questions in LLM control theory are further discussed in Section 6.

## 2 RELATED WORK

Much of the work on prompt optimization is concerned with finding prompts that induce higher LLM performance on "fill-in-the-blank" or "cloze" tasks (Taylor, 1953). One can frame a range of tasks including knowledge retrieval (Petroni et al., 2019), reasoning (Weston et al., 2016), and sentiment analysis (Wang et al., 2023b) as fill-in-the-blank tasks:

- **Knowledge Retrieval:** *"The Titanic sank in the year [MASK]."* (Answer: "1912")

- **Reasoning:** *"A is taller than B. B is taller than C. Is A taller than C? **Answer: [MASK]**"* (Answer: "Yes")

- **Sentiment Analysis:** *"I am sad today. **The sentiment of the previous sentence was [MASK]**"* (Answer: "Negative")

Notably, there is some freedom in the bolded "prompt text" that surrounds the question to convert it into a "fill-in-the-blank" task. As it turns out, the prompt tokens have a large effect on LLM performance (Brown et al., 2020; Zhang et al., 2022; Jiang et al., 2020).

Modern prompt optimization algorithms generally consist of two iterated steps: a sampling step where new prompts are generated and a testing step where the utility of the new prompts is evaluated, and the best are selected for the next iteration. Algorithms primarily differ in the sampling procedure, where various heuristics may be used to pick high-value swaps (Wen et al., 2023; Zhou et al., 2023;

Reynolds & McDonell, 2021). Overall, AutoPrompt and its derivative algorithms have been the most numerically successful prompt optimization methods, with the greedy coordinate gradient (GCG) algorithm having state-of-the-art performance (Zou et al., 2023).

**The AutoPrompt Family:** AutoPrompt (Shin et al., 2020) pioneered the current wave of prompt optimization. Shin *et al* propose a prompt optimization technique and demonstrate its effectiveness for engineering prompts to improve LLM performance on knowledge and sentiment analysis tasks. At its core, the AutoPrompt algorithm leverages gradient information at the token embedding layer to inform iterative token exchanges within the prompt. This method was extended in Zou et al. (2023) as the greedy coordinate gradient (GCG) algorithm. Taking inspiration from adversarial examples (Goodfellow et al., 2015), Zou *et al* applied this AutoPrompt variant to generate "jailbreak" prompts that cause aligned LLMs to generate objectionable content.

**Other Prompt Optimization Methods:** Other investigations on LLMs as prompt optimizers (Zhou et al., 2023) and further analysis of manual prompt optimization (Reynolds & McDonell, 2021) are informative but do not exceed the AutoPrompt family's performance. Some other methods include GBDA (Guo et al., 2021), an approach based on the Gumbel-Softmax reparametrization, the PEZ algorithm (Wen et al., 2023), which directly optimizes embeddings via gradient information, and FluentPrompt (Shi et al., 2022), which differs from AutoPrompt by incorporating Langevin dynamics. Despite the variety of alternatives, GCG retains state-of-the-art performance.

**Control Theory for LLMs:** To our knowledge, the only other work to date on the controllability of LLMs is Soatto et al. (2023). Soatto et al analyze the controllability of LLMs in terms of "meaningful sentences", defined as the sigma-algebra generated by snippets of text written on the Internet. Their empirical analysis revolves around demonstrating that LLMs are capable of attributing meaning. The theoretical analysis of LLM controllability is limited to "meaningful sentences", eliminating the possibility of out-of-distribution inputs and outputs. These restrictions render their results challenging to leverage toward a practical understanding of LLM controllability. We situate our work as a practically oriented exploration of LLM controllability. Motivated by challenges in developing LLM systems, we do not eliminate "meaningless sentences" from the state space or input space. We aim to establish a rigorous, general framework for understanding LLM systems and controllability that is amenable to the development of theory and practical engineering insights on systems design.

## 3 CONTROL THEORY FOR LLMS

Control theory originates from the study of automatic control systems in engineering. It seeks to understand how a "plant" system can be influenced toward a desired state using a "control signal" – often in the presence of disturbances and uncertainty.

Control theory is central to a variety of engineering problems, from electrical engineering to autopilot to telecommunications to manufacturing. Surprisingly, control theory has also been highly applicable to a diverse range of scientific disciplines. Analyzing systems through the lens of controllability has proven fruitful for generating insight into biological systems such as cell signaling pathways and neural networks (Yi et al., 2000), the economics of central banking (Anița et al., 2011), and controlling the spread of infectious diseases (Roy et al., 2009). One of the central benefits of studying systems via controllability is that a range of questions and problems naturally emerge from the framing: *when is control possible? What is the cost of control? How computationally intensive is control?* These questions are both practically useful and often lead to fundamental insights about the nature of the system in question.

To develop a control theory of LLMs, we provide fundamental abstract definitions of systems and control (Appendix A). We apply them to define LLM systems and outline specific canonical control concepts and problems such as controllability and reachability that arise naturally for LLM systems.

**Language Model Notation:** We denote a causal language model using $P_{LM}$. $P_{LM}$ maps from an ordered list of tokens from a vocabulary set $\mathcal{V}$ (e.g., $\mathbf{x} \in \mathcal{V}^n$) to the probability distribution over the next token $P_{LM}(x_{n+1}|\mathbf{x}) \in [0,1]^{|\mathcal{V}|}$. We use $\mathcal{V}^*$ to denote the set of all possible sequences

of any length composed of tokens from $\mathcal{V}$. The addition operator indicates the concatenation of token sequences. Bolded lowercase variables (e.g., $\mathbf{x} = [x^1, \ldots, x^n]$) denote token sequences while unbolded lowercase variables refer to individual tokens (e.g., $x \in \mathcal{V}$).

While LLMs are at times leveraged in a manner that masks the iterative aspects of generation, the reality is that token generation and externally imposed "control input" sequences are generated and processed sequentially, leading to non-trivial system dynamics. Several key differences remain between LLM-based systems and systems typically modeled through ordinary differential equations (ODEs), which have long been a cornerstone in the study of continuous-time dynamical systems:

1. **Discrete state and time:** LLM systems operate on sequences of discrete tokens over a discrete time set, in contrast to the continuous state spaces and time sets studied in classical control theory.

2. **Shift-and-Grow State Dynamics:** Whereas the system state in an ODE-based system has a fixed size over time, the system state $\mathbf{x}(t)$ for LLM systems grows as tokens are added to the state sequence.

3. **Mutual exclusion on control input token vs. generated token:** The LLM system state $\mathbf{x}(t)$ is written to one token at a time. The newest token is either drawn from the control input $u(t)$ or is generated by the LLM by sampling $x' \sim P_{LM}(x'|\mathbf{x}(t))$. This differs from traditional discrete stochastic systems, where the control sequence and internal dynamics generally affect the state synchronously.

We begin by rigorously defining LLM systems with user input, drawing from the abstract mathematical definition of a system (Definition 7).

**Definition 1** (LLM System with Control Input). *An autoregressive LLM system with control input $\Sigma = (\mathcal{V}, P_{LM})$ consists of:*

- $\mathcal{T} = \mathbb{N}$ : *The **time set** is the natural numbers.*

- $\mathcal{X} = \mathcal{V}^*$ : *The **state space** consists of all possible token sequences of any length drawn from $\mathcal{V}$. We denote the state at time $t$ as $\mathbf{x}(t) = [x^0(t), \ldots, x^t(t)]$.*

- $\mathcal{U} = \mathcal{V} \cup \varnothing$ : *The **input** takes values from the vocabulary set $\mathcal{V}$ or null.*

- $\phi : \mathcal{X} \times \mathcal{U} \times \mathcal{T}^2 \to \mathcal{X}$ : *The **transition map** is*

$$\phi(\mathbf{x}(t), u(t), t, t+1) = \begin{cases} \mathbf{x}(t) + u(t) & \text{if } u(t) \neq \varnothing \\ \mathbf{x}(t) + x' | x' \sim P_{LM}(x'|\mathbf{x}(t)) & \text{else} \end{cases} \tag{1}$$

  *Note that the general multi-step transition map $\phi(\mathbf{x}(t), u, t, t+N)$ can be achieved by iterating equation 1 for control sequences $u$ defined over the interval $[t, t+N]$.*

- $h(\mathbf{x}(t); r) = [x^{t-r}(t), \ldots, x^t(t)]$ : *The **readout map** returns the most recent $r$ tokens from state $\mathbf{x}(t)$.*

We note that this LLM system definition is generalizable to a variety of LLM augmentation, including chain-of-thought (Wei et al., 2023), retrieval-augmented generation (Lewis et al., 2020), and chatbot interaction. For example, chain-of-thought is equivalent to sampling the readout map $h(x(t), r)$ at time $T > |\mathbf{u}| + |\mathbf{x}_0| + r$ for prompt $\mathbf{u}$ and initial state $\mathbf{x}_0$. A similar formulation may be applied to LLM systems endowed with programmatic tools (e.g., Patil et al. (2023)).

In Definition 1, we assume that the control input gets to "decide" whether to yield token generation to the LLM ($u(t) = \varnothing$) or override the LLM and add some token $u(t) \neq \varnothing$ to the state $\mathbf{x}(t)$. This assumption generally holds when building LLM systems, though it may not hold when using existing systems (e.g., via non-streaming API). When discussing finite-length control inputs – e.g., the family of $k$-long input sequences $\mathbf{u} \in \mathcal{V}^k$ – the value of $u(\ell) : \ell > k$ is implicitly $\varnothing$ unless otherwise stated.

While next token generation $x' \sim P_{LM}(x'|\mathbf{x}(t))$ in equation 1 is probabilistic, we may render the system deterministic by sampling with zero temperature (i.e., greedy decoding). The greedy decoding assumption provides a foothold to analyze the reachable sets and controllability of LLM

systems without invoking notions of stochastic control as in Sivaramakrishnan et al. (2023); Soatto et al. (2023). Moreover, it remains connected to temperature-based stochastic decoding strategies as a limiting case of temperature-based sampling.

We now extend Definition 10 to define output controllability for LLM systems:

**Definition 2** (LLM Output Reachability). *Output token sequence* $\mathbf{y} \in \mathcal{V}^r$ *is reachable from initial state* $\mathbf{x}_0 \in \mathcal{V}^*$ *for LLM system* $\Sigma(\mathcal{V}, P_{LM})$ *iff there exists some time* $T$ *and input* $\mathbf{u}^* \in \mathcal{U}^T$ *that steers the LLM from initial state* $\mathbf{x}_0$ *to output* $\mathbf{y} = h(\mathbf{x}(T), r)$ *at time* $T$.

We disregard the trivial solution wherein the control input $\mathbf{u}^*(t)$ overrides the LLM to force the state sequence to take on the desired output value $\mathbf{y}$.

The reachable output set definition for LLM systems follows from Definition 11:

**Definition 3** (LLM Reachable Output Set). *The reachable output set from initial state* $\mathbf{x}_0 \in \mathcal{V}^*$ *for LLM system* $\Sigma = (\mathcal{V}, P_{LM})$ *is denoted* $R_y(\mathbf{x}_0)$ *and consists of all reachable outputs* $\mathbf{y} \in \mathcal{V}^*$ *from initial state* $\mathbf{x}_0$.

Output controllability for LLMs follows from Definition 13.

**Definition 4** (LLM Output Controllability). *An LLM system* $\Sigma = (\mathcal{V}, P_{LM})$ *is output controllable iff, for every initial state* $\mathbf{x}_0 \in \mathcal{V}^*$, *the reachable output set* $\mathcal{R}_y(\mathbf{x}_0) = \mathcal{V}^*$.

The turn-based nature of writing to the LLM state sequence $\mathbf{x}(t)$ invites the question of whether the prompt $\mathbf{u}$ should preempt the imposed state $\mathbf{x}_0$ or come after the state [1]. We focus our efforts on cases where $\mathbf{u}$ comes before imposed state sequence $\mathbf{x}_0$ due to its importance for developing system prompts and controlling text completion-based generation where the desired output is $\mathbf{x}_0 + \mathbf{y}^*$ for some desired continuation $\mathbf{y}*$ of partial string $\mathbf{x}_0$. Due to the costly nature of long prompts, we are especially interested in the existence of prompts $\mathbf{u}^*$ with minimal length $|\mathbf{u}^*|$.

Definitions 3 and 4 form the basis for our control theory of LLMs. While amenable to analytic analysis as in Section 4 and Soatto et al. (2023), empirical analysis of the reachable set and controllability is challenging due to the intractable size of $\mathcal{V}^*$. We propose the following statistical measure of controllability for practically assessing the controllability of an LLM system w.r.t. a dataset $\mathcal{D}$ under prompt length constraint $|\mathbf{u}| \leq k$:

**Definition 5** ($k - \epsilon$ Controllability). *Consider a dataset of state-output pairs* $\mathcal{D} = \{(x_0^i, y^i)\}_{i \in [N]}$. *An LLM* $\Sigma = (\mathcal{V}, P_{LM})$ *is* $k - \epsilon$ *controllable w.r.t.* $\mathcal{D}$ *if, for at least a proportion of* $(1 - \epsilon)$ *of* $(x_0^i, y^i) \in \mathcal{D}$, *the following condition holds:*

$$y^i \in \mathcal{R}_y^k(x_0^i) \tag{2}$$

*Where* $\mathcal{R}_y^k(x_0^i)$ *is the reachable set of outputs as in Definition 3 under the constraint that prompts* $\mathbf{u}$ *must have length* $|\mathbf{u}| \leq k$.

Our empirical work in Section 5.2 explores $k - \epsilon$ controllability w.r.t. initial states $\mathbf{x}_0$ sampled from the Wikitext dataset. While empirical analysis of LLM controllability is challenging due to the lack of apparent structure in LLM dynamics and the combinatorially large state space, we may still experimentally establish the *existence* of optimal prompts $\mathbf{u}^*$ that elicit a given output, and thus establish a lower bound on the content of the reachable set. Meanwhile, our theoretical work in Section 4 establishes upper bounds on the content of the reachable set for self-attention. We hope these complementary approaches may one day unify our understanding of LLM systems.

## 4 MATHEMATICAL ANALYSIS ON THE CONTROLLABILITY OF SELF-ATTENTION

Self-attention is a central component in modern transformer-based language models (Brown et al., 2020; Touvron et al., 2023; Radford et al., 2019; Min et al., 2023). Introduced in Vaswani et al.

---

[1] Both situations are reasonable in developing LLM systems: $\mathbf{u}$ preceding $\mathbf{x}_0$ may arise when prompting an LLM to complete a partial string $\mathbf{x}_0$. $\mathbf{u}$ proceeding $\mathbf{x}_0$ may arise when prompting an LLM in the presence of an imposed system prompt $\mathbf{x}_0$. Therefore, how an initial state $\mathbf{x}_0$ is interleaved with control input $\mathbf{u}$ is largely a design decision.

(2017), self-attention is the primary component in transformers where token representations exchange information.

**Definition 6** (Self-Attention). *Self-attention* $\Xi = (\mathbf{W_q}, \mathbf{W_k}, \mathbf{W_v})$ *is a map from* $\mathbb{R}^{N \times d_{in}} \to \mathbb{R}^{N \times d_{out}}$ *where* $N$ *is an arbitrary number of input token representations each of dimensionality* $d_{in}$, *and* $d_{out}$ *is the dimensionality of the output token representations.*

$$\Xi(\mathbf{X}) = \mathbb{D}^{-1} \exp\left(\frac{\mathbf{Q}\mathbf{K}^\top}{\sqrt{d_k}}\right) \mathbf{V} \tag{3}$$

*where* $\exp()$ *denotes element-wise exponentiation of the matrix entries,* $\mathbf{W_q}, \mathbf{W_k} \in \mathbb{R}^{d_{in} \times d_k}$, $\mathbf{W_v} \in \mathbb{R}^{d_{in} \times d_{out}}$, $\mathbf{Q} = \mathbf{X}\mathbf{W_q}$, $\mathbf{K} = \mathbf{X}\mathbf{W_k}$, $\mathbf{V} = \mathbf{X}\mathbf{W_v}$, *and* $\mathbb{D}$ *is a diagonal positive definite matrix defined as*

$$\mathbb{D} = diag\left(\exp\left(\frac{\mathbf{Q}\mathbf{K}^\top}{\sqrt{d_k}}\right) \mathbf{1}_{N \times 1}\right) \tag{4}$$

*where* $\mathbf{1}_{N \times 1}$ *is an* $N \times 1$ *matrix of ones.*

Note that the parameters and operation of $\Xi$ are independent of the number of token representations $N$. Self-attention may be applied to discrete token sequences $\mathbf{x} \in \mathcal{V}^*$ provided that each token $x^i \in \mathbf{x}$ is first mapped to a representation in the input dimension with some embedding map $\mathcal{E} : \mathcal{V} \to \mathbb{R}^{d_{in}}$.

We are interested in the reachability of output token representations, where we partition the input $\mathbf{X} \in \mathbb{R}^{(k+M) \times d_{in}}$ into an $k \times d_{in}$ block of control input representations $\mathbf{U}$ and an $M \times d_{in}$ block of imposed state representations $\mathbf{X}_0$ (cf. Definition 1). We also partition the output $\mathbf{X}' = \Xi(\mathbf{X}) \in \mathbb{R}^{(k+M) \times d_{in}}$ into a corresponding $k \times d_{out}$ matrix $\mathbf{U}'$ and an $M \times d_{out}$ matrix $\mathbf{Y}$. Motivated by the architecture of transformer-based language models, we seek to characterize the reachable set of output representations $\mathbf{Y} \in \mathcal{R}_y^k(\mathbf{X}_0)$ under imposed input representations $\mathbf{X}_0$ and controllable input representations $\mathbf{U}$, where $\mathbf{U}$ consists of $k$ token representations. While the reachable set is now a set of continuous-valued output representation matrices in $\mathbb{R}^{M \times d_{in}}$, we may readily adapt Definition 3 to define the reachable set for these conditions.

**Theorem 1** (Condition for Exclusion from the Reachable Set). *A desired output representation* $\mathbf{Y}^* \in \mathbb{R}^{M \times d_{out}}$ *must be excluded from the reachable set* $\mathcal{R}_y^k(\mathbf{X}_0)$ *if the following condition holds for any row* $i$:

$$\langle \mathbf{y}^{*i}, \hat{\mathbf{y}}_x^i \rangle \leq 0 \text{ and } \left\| \mathbf{y}^{*i} - \frac{\hat{D}_{xx}^i}{\hat{D}_{xx}^i + k \exp\left(\frac{\Omega_x \sigma_q \sigma_k \Omega_u}{\sqrt{d_k}}\right)} \hat{\mathbf{y}}_x^i \right\| \leq \sigma_q \Omega_u \tag{5}$$

*where* $\Omega_u = \max_j \|\mathbf{u}^j\|$ *for rows* $\mathbf{u}^j$ *of* $\mathbf{U}$, $\Omega_x = \max_j \|\mathbf{x}_0^j\|$ *for rows* $\mathbf{x}_0^i$ *of* $\mathbf{X}_0$, $\sigma_q$ *and* $\sigma_k$ *are the maximum singular values of* $\mathbf{W_q}, \mathbf{W_k}$ *respectively,* $\hat{D}_{xx}^i$ *is the* $i$th *element on the diagonal of* $\hat{\mathbf{D}}_{xx}$, *which is given by*

$$\hat{\mathbf{D}}_{xx} = diag\left(\exp\left(\frac{\mathbf{Q_x}\mathbf{K_x}^\top}{\sqrt{d_k}}\right) \mathbf{1}_{M \times 1}\right), \tag{6}$$

$\mathbf{y}^{*i}$ *is the* $i$th *row of* $\mathbf{Y}^*$, *and* $y_x^i$ *is the* $i$th *row of* $\hat{\mathbf{Y}}_x$, *which is given by*

$$\hat{\mathbf{Y}}_x = \hat{\mathbf{D}}_{xx}^{-1} \exp\left(\frac{\mathbf{Q_x}\mathbf{K_x}^\top}{\sqrt{d_k}}\right) \mathbf{V}_x \tag{7}$$

*where* $\mathbf{Q_x} = \mathbf{X_0}\mathbf{W_q}$, $\mathbf{K_x} = \mathbf{X_0}\mathbf{W_k}$, *and* $\mathbf{V_x} = \mathbf{X_0}\mathbf{W_v}$

*The proof of Equation 5 is provided in Appendix B.*

Intuitively, this condition arises when when the output representation $\hat{\mathbf{Y}}_x$ that occurs when only the imposed state $X_0$ is fed into the transformer is too far away for the control tokens $\mathbf{U}$ to steer it to $\mathbf{Y}^*$. The ability for the control input $\mathbf{U}$ to nullify the impact of $\hat{\mathbf{Y}}_x = \Xi(\mathbf{X}_0)$ on the output scales with the number of control input tokens $k$. A control input with many tokens can "dominate" the influence of $X_0$ by re-allocating attention away from the component of the output $\hat{\mathbf{Y}}_x$ that arises from $\mathbf{X}_0$. A notable insight from the proof is that one may decompose the output of attention into a components that arise largely from different parts of the input. While there are cross terms in the attention matrix, these amount to only a positive scaling factor applied to the "independent" components like $\hat{\mathbf{Y}}_x$. Thus, we have an analytic bound on the reachable output set for self-attention.

## 5 EXPERIMENTS

To gain a practical, empirical understanding of the reachable set $\mathcal{R}_y^k(\mathbf{x}_0)$, we probe the existence of optimal prompts $\mathbf{u}^*$ across datasets $\mathcal{D}$ of initial state–desired output pairs $(\mathbf{x}_0, y^*)$. We scope our experiments to study immediate control (i.e., we check the LLM output after $|y^*|$ tokens are generated) where the control input $\mathbf{u}$ is prepended to the imposed state $\mathbf{x}_0$. Moreover, we focus on the case of controlling the LLM system to produce a single output token $y^* \in \mathcal{V}$ under some constraint $|u| \leq k$. This "single-step" control renders the problem of gauging reachability computationally tractable and is a fundamental step toward understanding the iterated dynamics of LLM systems in terms of reachability and controllability. We leave the exploration of reachability and controllability under an extended time horizon (e.g., chain-of-thought, chatbot dynamics, tool-wielding LLMs) and under the requirement of multi-token outputs $\mathbf{y}$ to future work.

### 5.1 METHODS

We apply prompt optimization algorithms to establish the existence of optimal prompts $\mathbf{u}^*$ of length $k$ that steer the LLM system from initial state $\mathbf{x}_0$ to output $y$ for some dataset $\mathcal{D}$ of initial state-output pairs. In general, prompt optimization algorithms accept a token sequence and a loss function on said token sequence, along with a specification of which tokens are manipulable. The output of a prompt optimizer is a manipulated token sequence (i.e., optimized prompt) designed to minimize the loss. We apply two computational methods to generating optimal prompts: greedy back-generation (Algorithm C) and greedy coordinate gradient (GCG, Algorithm C, invented in Zou et al. (2023)). We found that greedy back-generation performed best for short prompts $k \leq 3$ tokens, while GCG was the best-performing algorithm for prompts of 4 or more tokens. To our knowledge, our greedy back-generation algorithm is novel. For brevity, we place the full description and our parameter values for the two algorithms in Appendix C, as the specifics of the algorithms are not the main contribution of this work.

We focus on understanding the content and structure of the reachable set of LLM system outputs $\mathcal{R}_y^k(\mathbf{x}_0)$, particularly under a constraint on the number of input tokens $k$. To prove which output tokens are reachable under varying input sequence lengths, we apply an incremental prompt lengthening procedure when searching for optimal prompts on some dataset $\mathcal{D}$.

---

**Algorithm 1** Back-off Prompt

---

**Require:** State-output token sequence $(\mathbf{x}_0, y)$; LLM system $\Sigma = (P_{LM}, \mathcal{V})$.
 1: **for** $k$ from 1 to 3 **do**
 2: $\quad$ $\mathbf{u}_k$ = Greedy Back Generate$(\mathbf{x}_0, y; \Sigma)$
 3: $\quad$ **return** $\mathbf{u}_k$ if it steers $\Sigma$ from $\mathbf{x}_0 \to y$.
 4: **end for**
 5: **for** $k \in [4, 6, 8, 10]$ **do**
 6: $\quad$ $\mathbf{u}_k$ = Greedy Coordinate Gradient$(\mathbf{x}_0, y; \Sigma)$
 7: $\quad$ **return** $\mathbf{u}_k$ if it steers $\Sigma$ from $\mathbf{x}_0 \to y$.
 8: **end for**
 9: **return** Failed to establish reachability.

---

### 5.2 RESULTS

Our results revolve around the reachable set $\mathcal{R}_y^k(\mathbf{x}_0)$ for state sequences sampled from the Wikitext dataset. We applied the same Back-off Prompt strategy (Algorithm 1) to determine $k-\epsilon$ controllability for all experiments, varying the specifics of the dataset $\mathcal{D}$ (cf 5). We established the reachability of the "ground truth" next token $y$ proceeding state token sequence $\mathbf{x}_0$ in Wikitext. In our tests on a dataset of 5000 state-output sequences with states of length $8 - 32$ tokens, we found that the true next token $y$ is reachable over 97% of the time with a prompt of length $k \leq 10$ (Figure 1).

To explore the reachable set $R_y^k(\mathbf{x}_0)$ beyond the ground truth of Wikitext outputs, we generated a synthetic dataset of outputs by sampling 25 Wikitext sequences $\mathbf{x}_0$ and selecting the top 75 most likely next-tokens according to the model itself $P_{LM}(y|\mathbf{x}_0)$ as the target tokens (Figures 1).

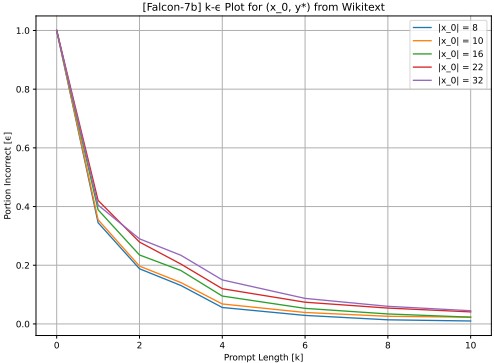 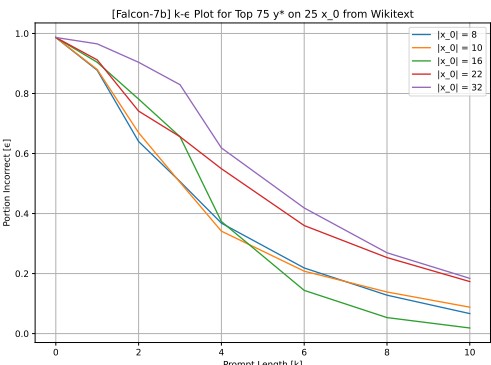

Figure 1: Main experimental results on $k - \epsilon$ controllability **Top left**: $k - \epsilon$ values for Falcon-7b on ground truth target token $y^*$. 97.16% of the instances were solved with a prompt of length $k \leq 10$.

**Top right**: $k - \epsilon$ values for reaching the top 75 most likely outputs $y^*$ for each $\mathbf{x}_0$. The top 75 targets were reachable at least 89.39% of the time with a prompt of length $k \leq 10$.

**Bottom right**: Prior likelihood rank of target token $y^*$ in terms of likelihood ascribed by the LLM $P_{LM}$ versus required prompt length to elicit $y^*$. Target tokens were sampled uniformly from the least to most likely token.

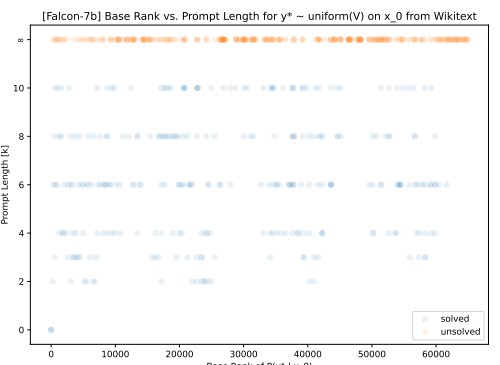

To maximally push the bounds of the reachable set within our single output token scope, we created another synthetic dataset where the target output token $y^*$ was sampled uniformly from the highest likelihood next token to the lowest likelihood token. Although the overall reachability score was low (only 46.43% reachable with $k = 10$), we were intrigued by the near-uniform relationship between prior token rank (based on $P_{LM}(y|\mathbf{x}_0)$) versus the required number of prompt tokens. Figure 1 plots the relationship between prior target token rank based on $P(y^*|\mathbf{x}_0)$ and the required prompt length $k$ to elicit the prompt. While over half were unreachable, the remaining reachable tokens appear uniformly distributed in terms of required prompt length, regardless of rank. We replicated these results (and more) across Llama-7b and Falcon-40b in Appendix D.

## 6 DISCUSSION

We proposed a control theoretic framework for understanding language model prompting, orienting our investigation around the reachable set of outputs $\mathcal{R}_y^k(\mathbf{x}_0)$. We proved a bound on the reachable set of outputs for self-attention in terms of the singular values of its weight matrices, and we established fundamental results on the reachability of "correct" next tokens (according to Wikitext). We expanded the scope of this investigation by probing the reachability of tokens assigned high likelihood by the LLM itself, and tokens assigned minimal likelihood by the LLM itself.

Bounding the reachable set for self-attention is deeply related to the mechanism by which consistent representations are formed for multi-token generation. Steering a language model to generate a desired token sequence requires that the control input induce a token representation in the right-most token such that the next token prediction logits $P(\mathbf{y}|\mathbf{u} + \mathbf{x}_0)$ achieves a desired value. Moreover, generated tokens are fed back into the model, and their representations must be steered as well to control iterated generation. Self-attention is the primary mechanism by which the token representations exchange information, making the reachable set of output representations across multiple tokens in $\mathbf{X}_0$ for self-attention a fundamental part of LLM control theory.

Our empirical results suggest that there is far more to the reachability of a given output than just prior likelihood or the prior rank the LLM assigns to a given token. Although prompt optimization-based $k - \epsilon$ controllability experiments are only able to provide a lower bound on the content of the reachable set, the ability to frequently control even the *least likely* token to being the *most likely* token with just a few input tokens is intriguing. We believe this result indicates the importance of further investigating the reachability and controllability of LLMs, particularly for developing capable and reliable LLM systems.

Our investigations provide an entry into the understanding of LLM controllability via prompts. However, a comprehensive understanding necessitates extending our exploration into diverse regimes. Exploring the controllability with longer prompts and longer questions (base token sequences) will be pivotal. Equally important is the study of diverse models to verify the generality of our findings. Moreover, the direct comparison of controllability scores of different model families is challenging since family uses a different tokenizer. The Llama family tokenizer, for instance, has a vocabulary of 30,000 tokens whereas the Falcon family has a vocabulary of 65,536 tokens. Further work is required to robustly compare controllability across models.

An intriguing observation from our study is the log-linear relationship between prompt length $k$ and controllability fraction $\epsilon$ (see Figure 2 in Appendix D). While this is compelling within our studied domain, it raises the essential question: is this relationship robust outside our current explorative scope? Unearthing universal scaling laws in LLM controllability would not only inform practical control applications but also open the door for theoretical insight into the nature of LLM behavior.

The progress we have made, both in understanding the bounds on self-attention controllability and the empirical measures of $k - \epsilon$ LLM controllability, underscores the potential of this control theoretic framing for studying LLMs. Below is a non-exhaustive list of open problems in LLM control, all stemming from the framing in section A:

- **Control Properties of Chain-of-Thought:** Chain-of-Thought is a powerful technique where LLMs are allowed to generate intermediate tokens (i.e., "thoughts") between a question and an answer (Wei et al., 2023). The control properties (e.g., stability, reachability) of systems leveraging these techniques are of great interest for understanding and composing systems of LLMs in the real world.

- **Distributional Control:** To what extent can we control the output distribution of a language model $P_{LM}(\mathbf{y}|\mathbf{x}_0 + \mathbf{u})$ to a desired distribution $P^*(\mathbf{y})$?

- **Computational Cost of Control:** What are the performance characteristics of LLM control regularized by computational cost?

- **Learnability of Control:** To what extent can LLMs learn to control each other? Work such as Zhou et al. (2023) showed that LLMs are capable of human-level prompt engineering, but it is unclear how well an LLM can learn to control another when explicitly optimized on the objective of LLM control.

- **Controllable Subspaces:** In the control of linear dynamical systems, it is known that uncontrollable systems are often coordinate transformable into a representation where a subset of the coordinates are controllable and a subset are uncontrollable Sontag (2013). We have shown that controllable and uncontrollable components naturally emerge for self-attention heads in section 4 – can this be generalized to transformer blocks with nonlinearities and residual streams?

- **Composable LLM Systems:** One of the greatest boons of control theory is the ability to compose control modules and subsystems into an interpretable, predictable, and effective whole (Lian et al., 2002). The composition of LLM systems (potentially with non-LLM control modules) is an exciting avenue for scaling super intelligent systems.

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

## A  ABSTRACT SYSTEMS AND CONTROL THEORY BACKGROUND

This section aims to provide an overview of fundamental control-theoretic concepts from an abstract perspective. We primarily draw from Sontag (2013); Kalman et al. (1969), and Ogata (2010).

Diverse definitions of "system" or "machine" exist in the literature, all representing the same core concept but varying in mathematical details. We offer the following high-level definition based on Sontag (2013):

**Definition 7** (System). *A "system" or "machine" $\Sigma = (\mathcal{T}, \mathcal{X}, \mathcal{U}, \phi)$ consists of:*

- *$\mathcal{T}$ : The **time set** along which system state evolves.*

- *$\mathcal{X}$ : The **state space**.*

- *$\mathcal{U}$ : The **input space**.*

- *$\phi : \mathcal{X} \times \mathcal{U} \times \mathcal{T}^2 \to \mathcal{X}$ : The **transition map**.*

*A system may also be equipped with an output space and readout map $(\mathcal{Y}, h)$:*

- *$\mathcal{Y}$ : The **output space**.*

- *$h : \mathcal{X} \times \mathcal{U} \times \mathcal{T} \to \mathcal{Y}$ : The **readout map**.*

In other words, at time $t \in \mathcal{T}$, the system's state takes on values $x \in \mathcal{X}$, and the control input takes values $u \in \mathcal{U}$. The system evolves over time with the transition map $\phi(x, u, t, t')$ that returns the new state value $x' \in \mathcal{X}$ at time $t' > t$. A system can also have a readout map $h(x, u, t)$ that produces the output value $y \in \mathcal{Y}$ given the current time, state, and input value. An input $u \in \mathcal{U}$ defined over interval $[t, t']$ may be said to *steer the system* $\Sigma = (\mathcal{T}, \mathcal{X}, \mathcal{U}, \phi)$ from state $x_0$ to state $x'$ if $x' = \phi(x_0, u, t, t')$. A wide variety of systems are expressible within this framework. E.g., we obtain discrete-time dynamical systems for $\mathcal{T} = \mathbb{Z}^+$. Continuous-time dynamical systems emerge for $\mathcal{T} = \mathbb{R}^+$.

Note that we assume that the system $\Sigma$ is time-invariant; its dynamics $\phi$ do not change as a function of time. This assumption is widely applicable and is often made in the literature (Kalman et al., 1969; Ogata, 2010; Sontag, 2013) to simplify definitions and discussions of systems.

Reachability is a core control theory concept and central to defining controllability. At their core, definitions of reachability revolve around the existence of control inputs $u \in \mathcal{U}$ that steer the system from a starting state $x_0 \in \mathcal{X}$ to some desired state(s). Following from Kalman et al. (1969); Sontag (2013), we define state reachability as:

**Definition 8** (State Reachability). *State $x \in \mathcal{X}$ is reachable from initial state $x_0 \in \mathcal{X}$ for system $\Sigma = (\mathcal{T}, \mathcal{X}, \mathcal{U}, \phi)$ iff there exists some time $T$ and control input $u^* \in \mathcal{U}$ such that $u^*$ steers the system from state $x_0$ to state $x$ at time $T$.*

We may use this definition of state reachability to define the reachable state set for some initial state $x_0 \in \mathcal{X}$:

**Definition 9** (Reachable State Set). *The reachable state set from initial state $x_0 \in \mathcal{X}$ for system $\Sigma = (\mathcal{T}, \mathcal{X}, \mathcal{U}, \phi)$ is denoted $\mathcal{R}(x_0) \subseteq \mathcal{X}$ and consists of all reachable states $x \in \mathcal{X}$ from initial state $x_0$ (cf. Definition 8).*

For systems with readout maps $h$, notions of *output reachability* arise naturally. Note that state reachability is neither necessary nor sufficient to guarantee output reachability.

**Definition 10** (Output Reachability). *Output $y \in \mathcal{Y}$ is reachable from initial state $x_0 \in \mathcal{X}$ for system $\Sigma = (\mathcal{T}, \mathcal{X}, \mathcal{U}, \phi, \mathcal{Y}, h)$ iff there exists some time $T$ and control input $u^* \in \mathcal{U}$ such that $u^*$ steers the system from state $x_0$ to output $y$ in time $T$.*

**Definition 11** (Reachable Output Set). *The reachable output set from initial state $x_0 \in \mathcal{X}$ for system $\Sigma = (\mathcal{T}, \mathcal{X}, \mathcal{U}, \phi, \mathcal{Y}, h)$ is denoted $\mathcal{R}_y(x_0)$ and consists of all reachable outputs $y \in \mathcal{Y}$ from initial state $x_0$ (cf. Definition 10).*

A system is controllable when the reachable set extends to the entire state space. Practically speaking, this implies that one can steer the system from any initial state to any desired state.

**Definition 12** (State Controllability). *System $\Sigma = (\mathcal{T}, \mathcal{X}, \mathcal{U}, \phi)$ is state controllable iff, for every initial state $x_0 \in \mathcal{X}$, the reachable set $\mathcal{R}(x_0) = \mathcal{X}$.*

**Definition 13** (Output Controllability). *System $\Sigma = (\mathcal{T}, \mathcal{X}, \mathcal{U}, \phi, \mathcal{Y}, h)$ is output controllable iff, for every initial state $x_0 \in \mathcal{X}$, the reachable output set $\mathcal{R}_y(x_0) = \mathcal{Y}$.*

A range of fruitful questions stem from these definitions: if there is a cost associated with control inputs $u \in \mathcal{U}$ (e.g., power constraints, length constraints), what is the minimum cost of control? What is the minimum time required to get from the initial state to the desired final state or output? If the system is not completely controllable, under what conditions is it controllable? Under which readout maps is a system output controllable?

# B    PROOF OF SELF-ATTENTION CONTROLLABILITY THEOREM 1

*Note: Key terms for the proof are introduced in Section 4 surrounding Theorem 1. Specifically, the definition of self-attention mechanism $\Xi$, the control problem setup, and the reachable set $\mathcal{R}_y^k(\mathbf{X}_0)$ are required background for this proof.*

*Proof.* For each token representation matrix $\mathbf{Q}, \mathbf{K}, \mathbf{V} \in \mathbb{R}^{(k+M)\times\cdot}$, we denote the first $k$ rows corresponding to $\mathbf{U}$ using $u$ as a subscript, like $\mathbf{Q}_u$. The remaining $M$ rows corresponding to $\mathbf{X}_0$ are denoted with subscript $x$ like $\mathbf{Q}_x$.

Let $\mathbf{A}$ be the exponentiated query-key outer product matrix with the following block structure:

$$\mathbf{A} = \exp\left(\frac{\mathbf{Q}\,\mathbf{K}^\top}{\sqrt{d_k}}\right) = \exp\left(\begin{bmatrix} \mathbf{Q_u K_u^\top} & \mathbf{Q_u K_x^\top} \\ \mathbf{Q_x K_u^\top} & \mathbf{Q_x K_x^\top} \end{bmatrix} \frac{1}{\sqrt{d_k}}\right) = \begin{bmatrix} \mathbf{A}_{uu} & \mathbf{A}_{ux} \\ \mathbf{A}_{xu} & \mathbf{A}_{xx} \end{bmatrix} \tag{8}$$

We apply a similar quadrant decomposition to $\mathbb{D}$, defined initially in Equation 4.

$$\mathbb{D} = \mathrm{diag}\left(\exp\left(\frac{\mathbf{Q}\mathbf{K}^\top}{\sqrt{d_k}}\right)\mathbf{1}_{N\times 1}\right) = \begin{bmatrix} \mathbb{D}_u & \mathbf{0} \\ \mathbf{0} & \mathbb{D}_x \end{bmatrix} \tag{9}$$

where the quadrant demarcations in $\mathbb{D}$ follow from Equation 8.

We may now express the self-attention mechanism output representations $\mathbf{Y}$ as

$$\mathbf{Y} = \mathbb{D}_x^{-1}\mathbf{A}_{xu}\mathbf{V}_u + \mathbb{D}_x^{-1}\mathbf{A}_{xx}\mathbf{V}_x \tag{10}$$

We begin by stating the equality between the desired output $\mathbf{Y}^*$ and the true system output from Equation 10. The final bound in Equation 5 of Theorem 1 is derived by isolating terms depending on control input $\mathbf{U}$, bounding them, and expressing that bound as a condition for achieving equality between the desired output $\hat{\mathbf{Y}}^*$ and the true system output.

$$\mathbf{Y}^* = \underbrace{\mathbb{D}_x^{-1}\mathbf{A}_{xu}\mathbf{V}_u}_{\triangleq \mathbf{Y}_u} + \underbrace{\mathbb{D}_x^{-1}\mathbf{A}_{xx}\mathbf{V}_x}_{\triangleq \mathbf{Y}_x} \tag{11}$$

$$\implies \mathbf{Y}_u = \mathbf{Y}^* - \mathbf{Y}_x \tag{12}$$

We may immediately bound the magnitude of the rows of $\mathbf{Y}_u$ as the matrix $\mathbb{D}_x^{-1}\mathbf{A}_{x_u}$ has rows that sum to less than 1 (it represents one quadrant of the row-wise softmaxed attention map, which has rows that sum to 1 by construction). Therefore, each row $\mathbf{y}_u^i$ of $\mathbf{Y}_u$ lies within the convex hull defined by the row vectors $\mathbf{v}_u^i$ of $\mathbf{V}_u$. Recalling Definition 6, $\mathbf{V}_u = \mathbf{U}\mathbf{W_v}$. Let $\Omega_u = \max_j \|\mathbf{u}^j\|$ for rows $\mathbf{u}^j$ of $\mathbf{U}$, we can bound the norm of each $\mathbf{v}_u^i$ in $\mathbf{V}_u$ with the maximum singular value of parameter matrix $\mathbf{W}_v$, denoted $\sigma_q$. Refer to Chapter 5 of Calafiore & El Ghaoui (2014) for an overview of singular values. Thus we may bound each $\|\mathbf{v}_u^i\| \le \Omega_u \sigma_q$. By the properties of convex hulls, each row of $\mathbf{Y}_u$ must inherit this upper bound on magnitude to retain feasibility.

$$\|\mathbf{y}_u^i\| < \Omega_u \sigma_q \tag{13}$$

Refer to Chapter 8 of Calafiore & El Ghaoui (2014)) for a detailed explanation of convex hulls and their properties.

While $\mathbf{Y}_x$ in Equation 11 may appear to depend only on imposed $\mathbf{X}_0$, the denominator term $\mathbb{D}_x^{-1}$ contains influences from $\mathbb{U}$. Let us split the denominator term $\mathbb{D}_x = \hat{\mathbf{D}}_{xx} + \hat{\mathbf{D}}_{xu}$ where $\hat{\mathbf{D}}_{xx}$ depends solely on the imposed input $\mathbb{X}_0$. $\hat{\mathbf{D}}_{xx}$ is definedin Equation 6. Let $\hat{\mathbf{D}}_{xu}$ be defined as:

$$\hat{\mathbf{D}}_{xu} = \mathrm{diag}\left(\exp\left(\frac{\mathbf{Q_x K_u}^\top}{\sqrt{d_k}}\right)\mathbf{1}_{k\times 1}\right) \tag{14}$$

Recall Equation 7, which defines $\hat{\mathbf{Y}}_x$, the output of $\Xi$ if only $\mathbf{X}_0$ is input. Let us express the condition in Equation 11 using $\hat{\mathbf{Y}}_x$ to disentangle the influence of the control input:

$$\mathbf{Y}_u = \mathbf{Y}^* - (\hat{\mathbf{D}}_{xu} + \hat{\mathbf{D}}_{xx})^{-1}\hat{\mathbf{D}}_{xx}\hat{\mathbf{Y}}_x \tag{15}$$

Observe that the rows of $\mathbf{Y}_x$ and $\hat{\mathbf{Y}}_x$ are positively scaled versions of each other because the denominator matrices are all positive and diagonal. Applying the bound in Equation 13 using row-wise notation,

$$\|\mathbf{y}^{*i} - \frac{\hat{D}^i_{xx}}{\hat{D}^i_{xx} + \hat{D}^i_{xu}} \hat{\mathbf{y}}^i_x\| \le \sigma_q \Omega_u \tag{16}$$

Using the same singular values reasoning as in Equation 13 to bound the unknown denominator term $\hat{D}^i_{xu}$, which is the only term still dependent on the control input $U$.

$$\hat{D}^i_{xu} \le k \exp\left(\frac{\Omega_x \sigma_q \sigma_k \Omega_u}{\sqrt{d_k}}\right) \tag{17}$$

Achieving this minimum value will minimize the value of $\mathbf{y}^i_x$ by maximally scaling down $\hat{\mathbf{y}}^i_x$. The maximum value for $\mathbf{y}^i_x$ arises when $\hat{D}^i_{xu}$ is minimized (e.g., to zero) resulting in $\mathbf{y}^i_x = \hat{\mathbf{y}}^i_x$.

Therefore, the the value of $\mathbf{y}^i_x$ is constrained linear scalings between this minimum and this maximum. If every scaling violates the inequality in Equation 16, then the system is strictly controllable.

Therefore, if $\langle \mathbf{y}^{*i}, \hat{\mathbf{y}}^i_x \rangle \le 0$ for some row $i$ following inequality is met, the output $Y^*$ is **strictly unreachable** under imposed input representations $\mathbf{X}_0$ and control input $\mathbf{U}$:

$$\|\mathbf{y}^{*i} - \frac{\hat{D}^i_{xx}}{\hat{D}^i_{xx} + k \exp\left(\frac{\Omega_x \sigma_q \sigma_k \Omega_u}{\sqrt{d_k}}\right)} \hat{\mathbf{y}}^i_x\| \le \sigma_q \Omega_u \tag{18}$$

$\square$

## C    PROMPT OPTIMIZATION ALGORITHMS

**Greedy Back-Generation:**    While testing all prompts in $\mathcal{V}^k$ is intractable for $k > 1$, it takes only $|\mathcal{V}|$ forward passes of the network to compute the loss on $y$ induced by all possible *single token* prompts $u \in \mathcal{V}$. Our Greedy Back Generation algorithm leverages this fact to generate prompts $u \in \mathcal{V}^k$ one token at a time, working backward sampling the $i$th greedy-optimal single token extension $u' = \arg\max_{u'} P_{LM}(y|u' + u + x)$ of the current prompt $u \in \mathcal{V}^{i-1}$.

---
**Algorithm 2** Greedy Token-Wise Prompt Generation
---
**Require:** A causal LLM $P_{LM}$ with vocabulary $\mathcal{V}$, a set of base tokens $x \in \mathcal{V}^n$, a desired final token $y \in \mathcal{V}$, and a desired number of prompt tokens $k$.
**Ensure:** *Magic words* $u^*$ of length $k$.
 1: Initialize $u^*$ to be empty.
 2: **for** $i$ from 1 to $k$ **do**
 3:     **for all** $u' \in \mathcal{V}$ **do**
 4:         compute $P_{LM}(y|u' + u^* + x)$
 5:     **end for**
 6:     Select the $u'$ that maximizes the probability of $y$ given $u' + u^* + x$. Prepend $u'$ to $u^*$
 7: **end for**
 8: **return** $u^*$
---

This method is optimal for $k = 1$ prompt token $u^* \in \mathcal{V}$ and generally outperforms GCG for short prompts of length $k \leq 3$. Computing 1 additional prompt token takes roughly 1-4 minute when using an NVIDIA A100-80GB GPU with a 7 billion parameter model and 5-20 minutes on 2 NVIDIA A100-80GB GPUs with a 40 billion parameter model.

**Greedy Coordinate Gradient (GCG):**    The Greedy Coordinate Gradient algorithm, presented by (Zou et al., 2023) building off the work of (Shin et al., 2020), is the state-of-the-art method for optimizing prompts. Starting with a random prompt of length $k$, the algorithm generates a batch of alternative prompts. Each member of the batch swaps a random token in the current prompt with a promising alternate token. The value metric for a swap is given by a first order approximation of the change in loss $\mathcal{L} = \text{CELoss}(y, P_{LM}(y|u + x))$ with the embedding of each token in $u$.

---
**Algorithm 3** Greedy Coordinate Gradient
---
**Require:** A causal LLM $P_{LM}$ that accepts token strings from a vocabulary $\mathcal{X}$, an embedding dictionary $\mathbf{e}$, embeddings $\mathbf{e}_i^*$ coroesponding to each token $i$ of $u^*$, a set of base tokens $x_{1:n}$, a desired number of prompt tokens $k$, iterations $T$, $k_{sub}$, and batch size $B$.
**Ensure:** *Magic words* $u^*$ of length $k$.
 1: Initialize $u^*$ to be random tokens from vocabulary.
 2: **for** $iteration$ from 1 to $T$ **do**
 3:     **for** $i$ from 1 to $k$ **do**                          ▷ Compute the top $k_{sub}$ most promising substitutions.
 4:         $\mathcal{X}_i = \text{Top-}k_{sub}(\mathbf{e}^T \nabla_{\mathbf{e}_i^*} P_{LM}(x_n|u^* + x_{1:n-1}))$
 5:     **end for**
 6:     **for** $b$ from 1 to $B$ **do**
 7:         $i = \text{randint}([1, \ldots, k])$                          ▷ Select random position to swap.
 8:         $j = \text{randint}([1, \ldots, k_{sub}]$                          ▷ Select random token from candidate set.
 9:         $\tilde{u}_b^*[i] = \mathcal{X}_i[j]$                          ▷ Swap token at position $i$.
10:     **end for**
11:     $u^* = \tilde{u}_{b^*}^*$ , where $b^* = \text{argmax}_b(P_{LM}(x_n|u^* + x_{1:n-1})))$          ▷ Select replacement which maximizes probability of future token.
12: **end for**
13: **return** $u^*$
---

This method outperforms all other methods we tested for prompts of length $k > 3$. We use a batch size $B = 768$, sampled from the top $k_{sub} = 128$ token replacements at each index, and iterate for $T = 34$ iterations. For each instance, this optimization took roughly 2 minutes for the 7 billion parameter models on a single A100-80GB GPU and 4-8 minutes for the 40 billion parameter model on 4 A100-80GB GPU.

# D    SUPPLEMENTARY FIGURES: OPTIMAL CONTROL PROMPTS

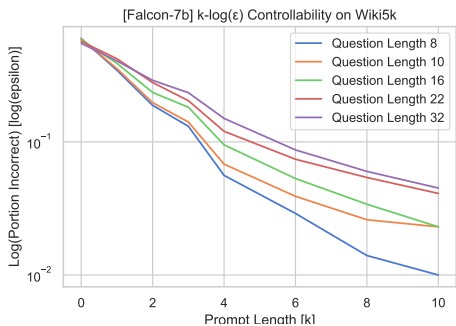

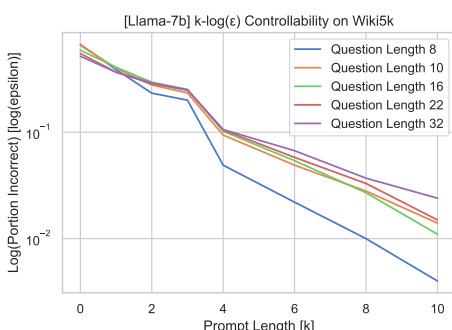

Figure 2: Log spaced main results of $k-\log(\epsilon)$ controllability. Interestingly, the relationship between $k$ and $\log(\epsilon)$ appears roughly linear for each question length in the regime studied.
**Top left**: $k - \log(\epsilon)$ values for Falcon-7b.
**Top right**: $k - \log(\epsilon)$ values for Llama-7b.
**Bottom right**: $k-\log(\epsilon)$ values for Falcon-40b.

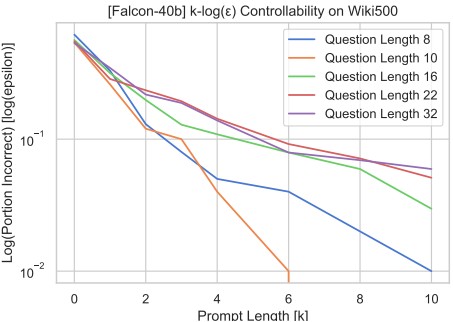

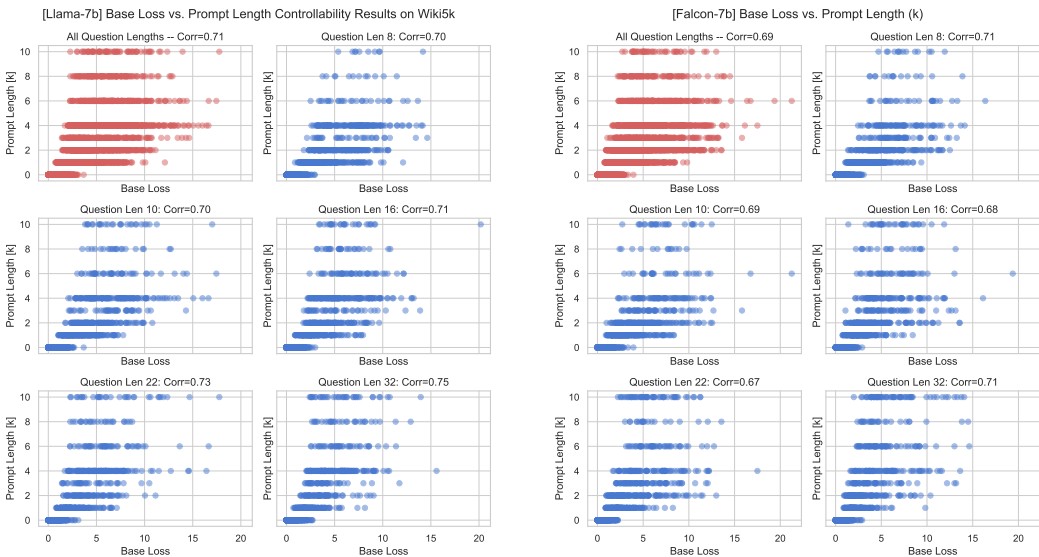

Figure 3: Prompt length $k$ versus base cross-entropy loss on the final token for Llama-7b.

Figure 4: Prompt length $k$ versus base cross-entropy loss on the final token for Falcon-7b.

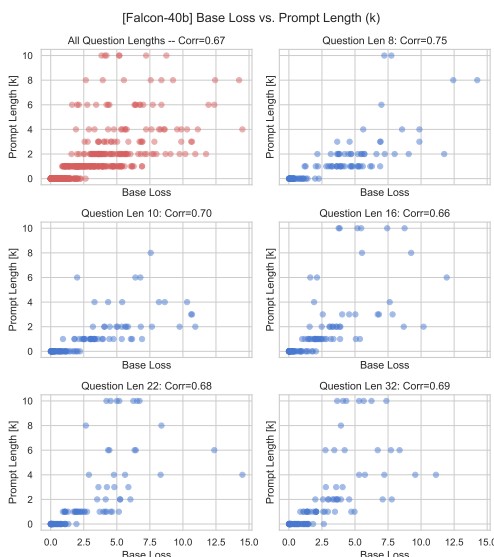

Figure 5: Prompt length $k$ versus base cross-entropy loss on the final token for Falcon-40b.

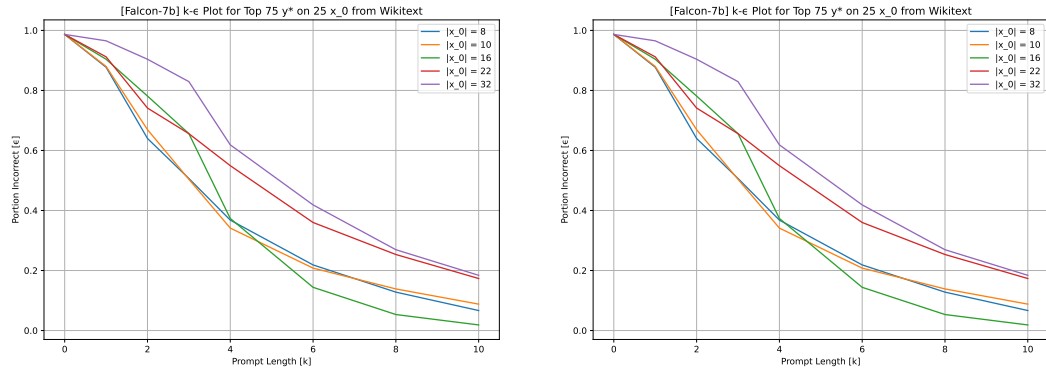

Figure 6: Llama-7b $k - \epsilon$ reachability on top-75 data.

Figure 7: Falcon-7b $k - \epsilon$ reachability on top-75 data.

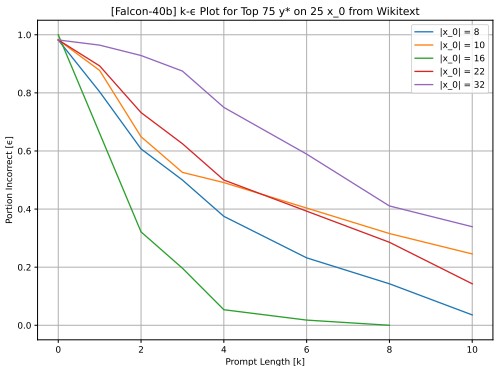

Figure 8: Falcon-40b $k - \epsilon$ reachability on top-75 data.

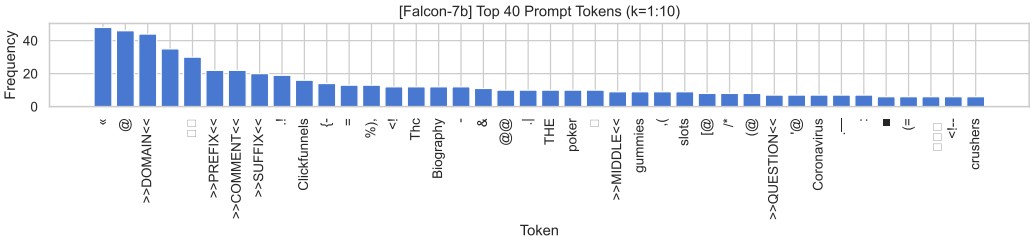

Figure 9: Prompt token frequencies for Falcon-7b across Wikitext5k instances for $k = 1 : 10$.

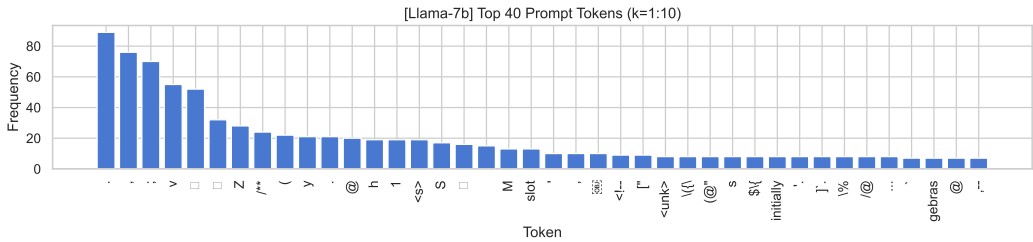

Figure 10: Prompt token frequencies for Llama-7b across Wikitext5k instances for $k = 1 : 10$.

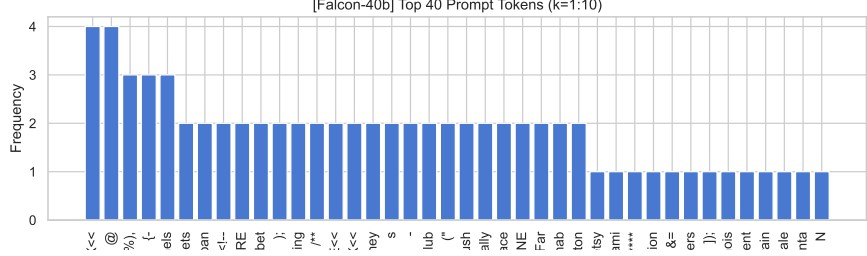

Figure 11: Prompt token frequencies for Falcon-40b across Wikitext500 instances for $k = 1 : 10$.

