# OpenReview forum: "What's the Magic Word? A Control Theory of LLM Prompting"
_ICLR.cc/2024/Conference — Submitted to ICLR 2024_

### Official Review · Reviewer_EeUn · 2023-10-17

**Soundness:** 2 fair
**Presentation:** 2 fair
**Contribution:** 2 fair
**Rating:** 3
**Confidence:** 3

**Summary:**

The paper suggests to view prompt engineering through the lens of control theory. The authors define a notion of probabilistically approximate controllability and verify study this property in various language models empircally. They also provide a sort of theoretical "converse", showing that if a certain condition is met, the language model cannot be controllable as defined here.

**Strengths:**

- I think the papers main selling point is to offer the perspective to analyze prompt engineering through a control theory lens. Unfortunately, it not novel and has been previously suggested in [1].

- Nevertheless, the idea to use control theory---and notions of controllability/reachability in particular---to analyze LLMs is quite elegant.

- The paper asks a number of interesting questions in section 7.

[1] Soatto, Stefano, et al. "Taming AI Bots: Controllability of Neural States in Large Language Models." arXiv preprint arXiv:2305.18449 (2023).

**Weaknesses:**

- It is not clear that the analysis in section 4 is of any relevance as it stands for two reasons:
 1)  I am not sure it makes sense to give a "topological/norm/metric" controllability condition as in (4). Is there a natural topology here to justify this? As far as I understand tokenization imposes a more or less arbitrary choice of topology so it is not clear to me that characterizing controllability (an algebraic concept) by a topological one makes any sense.
2) Moreover, I cannot find any suggestion in the paper that the bound is of any practical use---here the obvious questions is: is it sharp and if not how loose?

- Control theory is fundamentally a study of dynamical systems. The statement that " For simplicity, the time set T and the transition map ϕ are omitted from this
definition" made in section 3 then suggests that you are actually no longer really in the realm of control---which was your main selling point to begin with.


- The experiments are somewhat sparringly commented and the reader is left wondering how these were conducted. What is the precise definition of a "solved instance" and how were these instances generated? My critique pertains to section 6 mainly but the appendix also suffers from uncommented plots.

- While satisfactory, the level of writing could be better. There are a number of grammatical errors, but beyond that and more importantly, the paper does not feel very well structured. I think this is due in part to a lack of clear delineation of what the paper's contribution is.

- It would be better if the analysis was stated in standard thm/proof style. Currently, commentary is interwoven with the proof of claim (4) making the derivation unnecessarily obtuse. Ideally, the proof should be prefaced by exactly that which is to be shown and then broken down into an overview of the relevant components, followed by the proof itself. Let me finally say that I find this particularly strange since the paper has a number of definitions (potentially too many...) stated in the standard mathematical style.

**Questions:**

- Is the norm in (4) the Euclidean norm?



See also questions I raised in weaknesses above.

---

> ### Author Response · Authors · 2023-11-29
> **RE: Reviewer EeUn [1/2]**
>
> We thank reviewer EeUn for their insightful input. Here is how we have addressed the points raised:
>
> **Prior use of control theory for LLMs**: We added the following acknowledgement of Soatto et al’s work on LLM control theory:
>
> “To our knowledge, the only other work to date on the controllability of LLMs is Soatto et al. (2023). Soatto et al analyze the controllability of LLMs in terms of “meaningful sentences”, defined as the sigma-algebra generated by snippets of text written on the Internet. Their empirical analysis revolves around demonstrating that LLMs are capable of attributing meaning. The theoretical analysis of LLM controllability is limited to “meaningful sentences”, eliminating the possibility of out-of-distribution inputs and outputs. These restrictions render their results challenging to leverage toward a practical understanding of LLM controllability. We situate our work as a practically oriented exploration of LLM controllability. Motivated by challenges in developing LLM systems, we do not eliminate “meaningless sentences” from the state space or input space. We aim to establish a rigorous, general framework for understanding LLM systems and controllability that is amenable to the development of theory and practical engineering insights on systems design.”
>
> **Relevance of Section 4 (Self-Attention Proof)**: The point raised about whether there is a “natural topology” to justify our analysis is an interesting one. We have added the following discussion in Section 6 thanks to reviewer EeUn’s insightful comment:
>
> “Bounding the reachable set for self-attention is deeply related to the mechanism by which consistent representations are formed for multi-token generation.
>
> Steering a language model to generate a desired token sequence requires that the control input induce a token representation in the right-most token such that the next token prediction logits P (y∣u + x0 ) achieves a desired value.
>
> Moreover, generated tokens are fed back into the model, and their representations must be steered to control iterated generation.
>
> Self-attention is the primary mechanism by which the token representations exchange information, making the reachable set of output representations across multiple tokens in X0 for self-attention a fundamental part of LLM control theory.”
>
> While tokenization at the input may impose an arbitrary choice of topology, the controllability of logits on the output side is smooth. We are interested in manipulating the logit vector to maximize the probability of some desired sequence. Characterizing conditions for reaching outputs in this space aligns well with our analysis. That said, we welcome any insight into other ways to characterize the controllability of self-attention, particularly in the context of token embeddings with an arbitrary topology.
>
> We have also updated the proof to be in a theorem environment and have clarified the relationship between token representations in self-attention and tokens in LLM systems in Section 4.
>
> **Dynamical vs. Non-Dynamical Systems**: We have revised our framework in light of input from EeUn and other reviewers on the missing dynamical aspect of LLM systems. Particularly relevant is Definition 1, which is our new definition for LLM systems, and our commentary at the beginning of Section 3 clarifying the relationship differences between LLM systems and more conventional dynamical systems:
>
> “While LLMs are at times leveraged in a manner that masks the iterative aspects of generation, the reality is that token generation and externally imposed ``control input'' sequences are generated and processed sequentially, leading to non-trivial system dynamics.
>
> Several key differences remain between LLM-based systems and systems typically modeled through ordinary differential equations (ODEs), which have long been a cornerstone in the study of continuous-time dynamical systems:
> 1. **Discrete state and time**: LLM systems operate on sequences of discrete tokens over a discrete time set, in contrast to the continuous state spaces and time sets studied in classical control theory.
> 2. **Shift-and-Grow State Dynamics**: Whereas the system state in an ODE-based system has a fixed size over time, the system state x(t) for LLM systems grows as tokens are added to the state sequence.
> 3. **Mutual exclusion on control input token vs. generated token**: The LLM system state x(t) is written to one token at a time. The newest token is either drawn from the control input u(t) or is generated by the LLM by sampling x′ ∼ PLM (x′∣x(t)). This differs from traditional discrete stochastic systems, where the control sequence and internal dynamics generally affect the state synchronously.”

---

> > ### Author Response · Authors · 2023-11-29
> > **RE: Reviewer EeUn [2/2]**
> >
> > **Clarity/experimental design**: We hope our new unified terminology across our theoretical and experimental contributions clarifies how our experiments were conducted. To further clarify our experiments, we included pseudocode for the “**Back- off Prompt**” algorithm that is used across the k − ε controllability experiments in Section 5. We appreciate the feedback on organization, and we hope our new theorem/proof style in Section 4 aids in communicating.
> >
> > **Euclidean norm**: Yes, the norms in the paper are all Euclidean.
> >
> > Once again, we would like to express our gratitude for your time and effort in reviewing our paper. Your feedback has been instrumental in enhancing our research, and we hope that the revised version addresses the concerns raised.

---

### Official Review · Reviewer_fWP6 · 2023-10-24

**Soundness:** 2 fair
**Presentation:** 2 fair
**Contribution:** 3 good
**Rating:** 5
**Confidence:** 3

**Summary:**

This work presents a control perspective of the LLM steerability by introducing the concept of $k-\epsilon$ controllability. They also proved a bound on the controllability of self-attention in terms of the singular values of its weight matrices. In addition, several experimental studies have performed to compute the $k-\epsilon$ controllability of LLMs (Falcon-7b, Llama-7b, Falcon-40b). The authors conclude that LLMs are very controllable and the control prompts of 10 tokens or less are sufficiently enough to ensure the LLM output the target token.

**Strengths:**

The idea of introducing $k-\epsilon$ controllability is interesting and it is a nice connection between LLM and control theory. The theoretical bound presented in Section 4, though hard to check in reality, is an initial step towards understanding the steerability of LLM theoretically. In addition, I do appreciate the authors provide an interesting discussion on the open problems of LLM in control.

**Weaknesses:**

Section 4 only considers a self-attention head, which is quite simple and limited (compared to the current model used in LLM). What are the difficulties in generalizing such results to a more complex model?

The presentation of Section 4 can be further improved. The relationship between state controllability (Definition 7) and $k-\epsilon$ controllability (Definition 6) should be discussed, i.e., implications of your theory result in Section 4. In addition, I am confused about some notations: are $u_i$, $x_i$ the embeddings of the tokens? Previously the u and x are presented as tokens, it does not make sense to make $\|u_i\|\le 1$ and $\|x_i\| \le 1$ if they are tokens. In addition, is this assumption valid in real LLMs?


Although introducing the controllability of LLM from a control perspective is interesting, the experimental results of checking the controllability of the LLMs are not very exciting given the existing results from previous work [Zou 2023]. The experiment setup is almost identical to GCG work and the obtained results are also within expectation. Instead, proposing a new method to study the controllability of black-box LLMs will be more interesting.

**Questions:**

1. What are the connections between state controllability (Definition 7) and $k-\epsilon$ controllability (Definition 6)? Does the former imply the later?
2. Are the assumptions $\| u_i\| \le 1$ and $\| x_i \| \le 1$ realistic? If not, is your results in Section 4 still hold?

---

> ### Author Response · Authors · 2023-11-29
> **RE: Reviewer fWP6**
>
> We thank reviewer fWP6 for their support and insightful comments.
>
> **Generalizing self-attention results**: We hope the revised theorem/proof structure addresses reviewer fWP6’s concerns on presentation. We hope that the reviewers find **paragraph 2 of the Discussion section** insightful as to how/why the analytic result on self-attention might be generalizable:
>
> “Bounding the reachable set for self-attention is deeply related to the mechanism by which consistent representations are formed for multi-token generation.
>
> Steering a language model to generate a desired token sequence requires that the control input induce a token representation in the right-most token such that the next token prediction logits P (y∣u + x0 ) achieves a desired value.
> Moreover, generated tokens are fed back into the model, and their representations must be steered to control iterated generation.
>
> Self-attention is the primary mechanism by which the token representations exchange information, making the reachable set of output representations across multiple tokens in $\mathbf X_0$ for self-attention a fundamental part of LLM control theory.”
>
> We have clarified that the self-attention mechanism deals in token representations, and we have removed the unit magnitude ∣ui∣ ≤ 1 condition in favor of a generic maximum magnitude value that is propagated through our results ∣ui∣ ≤ Ωu, where Ωu may be any positive number. Ωu is largely a free parameter that can be adjusted as needed depending on input normalization or other mechanisms that may influence the input representations. Overall, we believe that the central result of demonstrating a limit to the reachable outputs of self-attention as a function of a controllable input representation and an imposed state representation is an exciting way to approach understanding the controllability of LLMs from a theoretical/first-principles perspective, and we thank reviewer fWP6 for their feedback for improving this part of our work.
>
> **Experimental Results not Exciting**: We have expanded the scope of our experimental results to probe the reachability of a larger space of tokens: in particular, we explore the reachability of outputs y based on the prior probability PLM (y∣x0) for some imposed initial state x0. We establish that the top 75 most likely next tokens, as estimated by the LLM itself, are reachable at least 85% of the time with prompts of k ≤ 10 tokens. Intriguingly, short prompt sequences can dramatically alter the likelihood of specific outputs, even making the least likely tokens become the most likely ones. The bottom right plot in Figure 1 shows a practically uniform relationship between how highly the LLM ranked a given output token y given x_0 and how likely we were to find a control input u — even a short one — that forced output token y to the top of the probability distribution. We hope this expanded scope underscores the potential of this control- theoretic framing for studying LLMs.
>
> **Connection between state controllability and k − ε**: In light of other reviewer’s advice on improving the formalization, we have unified our definitions of control in terms of the reachable set Ry(x0). Please see Section 3 and the supporting material in Appendix A for our new, more cohesive definition.

---

### Official Review · Reviewer_8q5r · 2023-10-31

**Soundness:** 3 good
**Presentation:** 2 fair
**Contribution:** 2 fair
**Rating:** 5
**Confidence:** 3

**Summary:**

The paper presents a control theoretic perspective on steering LLMs via prompting. After formalizing the problem and providing a theoretical analysis of the conditions when self-attention is able unable to be steered, the paper provides an empirical investigation of the steerability of LLMs in practice on Wikitext.

**Strengths:**

- The paper presents a compelling perspective, which could be fruitful.
- Despite being a little bit disorganized, I appreciate the basic idea behind the theoretical analysis, showing that there is some fundamental bottleneck that makes LLMs not arbitrarily steerable (under some assumptions).

**Weaknesses:**

- I have concerns on the significance of the empirical results. In particular, I suspect LLMs might be very easily steerable, and that any limitation in the ability to push them to a particular output is just due to limitations in the optimization method that is used to find the prompt. As a noticeable example, one might say that, for the definition of steering that has been employed in the paper, a sufficiently capable LLM can always be steered by prepending to it an prefix that reads like "after reading n words/token, output this particular word/token". If the optimization procedure does not find such a solution, it seems more a limitation of that than a fact related to the inability of LLMs to be steered.
- The paper is at times not well-organized. The discussion section, that is usually a summary of the takeaways from the paper, looks more like a discussion of related work, and also has some incomplete points, the theoretical results would be better understood inside of a theorem latex environment, and so on.

**Questions:**

- How is your definition / empirical analysis entangled with the prompt optimization technique that is being used?
- Can you tidy up some parts of the paper to make them clearer to a machine learning audience?

---

> ### Author Response · Authors · 2023-11-29
> **RE: Reviewer 8q5r**
>
> Hi 8q5r,
>
> Thank you for your attentive feedback. We appreciate your critiques! To your point about prepending a prefix like “after reading n words/tokens, output this particular word/token”—your intuition is exactly where ours was initially. Your suggestion can be made even stronger/more effective by straight repetition. Say we had “The brown fox jumped over the lazy ____“, and wanted to fill in the blank with ‘dog’. Just prefix:
>
> “The brown fox jumped over the lazy dog The quick brown fox jumped over the lazy dog The quick brown fox jumped over the lazy dog...”
>
> repeated enough times to guarantee the correct output. However, the problem with this strategy is that, to work realistically, it requires a large number of prefix tokens. Imagine we had a ‘budget’ and could only prepend 4 tokens as our control prompt. Can you think of an analogous noticeable example for 4 tokens? What about 2? We find there is some prompt length (in our paper we call it k), after which an instance is no longer controllable.In fact, our k-epsilon empirical results systematically investigate the relationship between prompt length and controllability. Epsilon represents the proportion of instances for which we didn’t find a ‘magic word’ (a control prompt steering the model to the correct final token). In Figure 1, you can see how epsilon drops significantly as the prompt length increases. Longer prompts are more controllable! As for how our definition and empirical analysis is entangled with the prompt optimization technique, we state in our discussion that our methods provide a lower bound for the k-epsilon controllability of Falcon-7b, Falcon-40b, and Llama-7b. Notably, this bound is met for k=1, since our technique ‘greedy back generation’ tests all possible single-token prompts and finds the best one. For k > 1, because it is impossible to explore the exponentially growing prompt space completely, there may be magic words we have not found. However, because we actually find each magic word (though there could be some we don’t find), the models are at least as controllable as we report. As for organization, we uploaded a new revision fixing many of the organizational issues. We hope you find it more acceptable. Again, we appreciate your feedback and concerns raised.

---

### Official Review · Reviewer_jCxR · 2023-11-06

**Soundness:** 3 good
**Presentation:** 3 good
**Contribution:** 2 fair
**Rating:** 5
**Confidence:** 4

**Summary:**

This paper studies prompt engineering, a key factor in exploiting LLM, from a point of view in control theory. The main contribution is that:
1) Formulate the prompt engineering problem as an optimal control problem (e.g., defining the k-epsilon controllability). This provides a mathematical framework to study this problem rigorously.
2) Prove that if a certain eigenvalue bound about the weight matrices is satisfied, a single attention head is state controllable. That is, for any input, there exists a prompt, i.e., the "magic word", that can force the LLM to give the desired output.
3) Proposed two prompt searching algorithms to validate the controllability of Falcon-7b, Llama-7b, and Falcon-40b. Experiment results show that there generally exists a magic word of length 10 or less for over 97% of Wikitext dataset.

**Strengths:**

The primary novelty of this paper lies in its formulation of the prompt engineering problem as an optimal control problem.  I truly appreciate this idea. It is widely acknowledged that the choice of prompts heavily influences the LLM performance. The interplay between the LLM's weights and the input prompt jointly determines the "states" of the LLM. This insight is intuitive but non-trivial, very different from conventional supervised learning approaches. It is commendable that the authors have translated this observation into a mathematical framework and provided an initial analysis.

**Weaknesses:**

Weakness or Questions

1. I have a concern regarding the controllability metric. When an LLM is controllable, there exists a prompt capable of compelling the LLM to produce a desired output, even if that output is factually incorrect. Thus, this formulation does not seem to establish a real connection with the specific capabilities of LLM, such as reasoning and knowledge memorization. Moreover, The attribute of controllability appears to lean more towards a negative property, signifying the LLM's susceptibility to prompt manipulation and potential vulnerabilities.

2. The theory only requires the weight matrices to meet a particular bound on their largest eigenvalues. The analysis framework does not depend on any training data or training algorithms. Even randomly generated weights (as long as they satisfy the specific bound) can render the LLM entirely controllable. This raises concerns about whether this theoretical framework can effectively explain the underlying mechanism of a well-trained LLM.

3. The results do not seem to offer practical guidance for prompt design. The analysis does not appear to provide insights into how to stimulate the capabilities of an LLM effectively. The proposed methods both require access to the desired output (the ground truth) during the search process.

**Questions:**

My concerns and questions are merged into the "weakness" section. Please refer to the last section.

---

> ### Author Response · Authors · 2023-11-29
> **RE: Reviewer jCxR**
>
> Hi jCxR,
>
> Thank you for your time and consideration of our paper. Your concerns are well taken— from the perspective of an LLM engineer, a system designer, or a user, you are absolutely correct: we do not offer practical guidance. Instead, we take the perspective of scientists, treating LLMs as our object of study. We consider the LLM as a mathematical object for which the controllability can be determined.
>
> From this point of view, it was never our goal to improve factual correctness or the real abilities of LLMs, such as knowledge and reasoning. We are completely agnostic toward whether controllability is ‘good’ or ‘bad’, since we only aim to characterize and study it objectively and rigorously. Likewise, providing insights on prompt design or ways to stimulate LLM capabilities is outside of our scope. While it’s entirely possible that further research may use our rigorous foundation as a springboard for useful methodologies, our goal is strictly the objective scientific study.
>
> Your point about randomly generated weights is an astute observation. According to our analytical analysis, it is indeed the case that randomly generated weights are, at least in principle, as controllable as a fully trained network. Moreover, this result stems only from the mathematical structure of self-attention. Note that since self-attention is the only mechanism by which tokens can influence each other in an LLM, it is perhaps not so unexpected that the structure of self-attention should primarily characterize controllability.
>
> Again, we appreciate your feedback and comments, and have submitted a new revision addressing several issues. We hope you find it more acceptable!

---

### Official Review · Reviewer_4Qb8 · 2023-11-06

**Soundness:** 3 good
**Presentation:** 2 fair
**Contribution:** 3 good
**Rating:** 5
**Confidence:** 5

**Summary:**

This paper uses control theoretical concepts to analyze the existence of specific prompts (magic words) that allow for controllable text generation. The authors do a thorough review of the state-of-the-art literature on the topic, and introduce the necessary control theoretical concepts. Building on these concepts, the authors introduce the definition of k-\epsilon controllability, which is a measure of the existence of a “magic word” (k token prompt) that would lead to the desired output token. The authors present a theoretical result to bound the k-epsilon controllability of the attention head, and two heuristic algorithms that allow to compute magic words. They provide simulation results that show that in 97% of the cases surveyed, “magic words” exist.

**Strengths:**

This paper is one of the first of its kind in applying a control theoretical concepts to the study of LLMs. Control theory offers very powerful tools that have the potential of providing a more formal understanding of LLMs behavior. For this reason, the approach introduced in this paper has great potential to advance our understanding of language technologies. This reviewer positively values the originality of this paper. Moreover, the topic addressed of controlling language generation with the appropriate prompt is very relevant, and having rigorous tools opens the door to a formal treatment of LLMs. In this paper, they provide an algorithm to find “magic words” based on a mathematical result inspired by control theory concepts. More importantly, they list a series of open questions that could be addressed from a control-theoretical point of view.

**Weaknesses:**

This paper has some issues with the formalization of the theoretical concepts, as well as with the presentation of the results. Since this paper introduces for the first time control theoretical concepts in the light of LLMs, it is paramount that the definitions are accurate and properly capture the ideas underpinning dynamical systems. This is not the case for this paper. The control theoretical concepts in this paper are not properly communicated, and Definition 4 has several flaws. The definition given corresponds to an input-output system, as opposed to a dynamical system: in order for it to be a dynamical system, the state space should be V as opposed to V^*. Moreover, the definition given for k-\epsilon controllability is very far the definition of controllability in dynamical systems. Please refer to Feedback Control Systems (Amstrom and Murray, 2009) for details. This reviewer is concerned that, if published in the current form, this paper can introduce more confusion than clarifications in the realm of using control theoretical tools for LLMs analysis. Aside from this, the main result should be framed as a theorem and provided in the body of the paper, together with the algorithms, not in the appendices, as these are the main results of the paper. Furthermore, the paper currently lacks necessary formal definitions (such as the definition of the V^* set), and the control theoretical section lacks clarity in the exposition. Substantial modifications are needed to improve the rigor of the presentation in Sections 3 and 4.

**Questions:**

This reviewer would like to know why the state space is defined as V^* as opposed to V. Defining it in such manner strips away the dynamics of the system (progression with time), and reduces it to an input/output system. This reviewer would also like to know how map \phi is defined in the context of LLMs, since only h was defined. Since the state space is defined as V^*, the map \phi cannot properly be defined as the dynamics of the system. Moreover, the question under study is more a question of system excitation under an initial condition than a control input. In the context of control theory, an input is very rarely seen as a one-time excitation, but rather as a feedback signal. The setup described in this paper corresponds to an input/output behavior under an initial condition, and as a result, the definition for k-\epsilon controllability is very different from the standard controllability definition used in control theory. Moreover, this reviewer suggest the use of “reachability”, as opposed to “controllability”, as it more accurately captures the problem under consideration.

**Details Of Ethics Concerns:**

This reviewer does not have ethical concerns with this paper.

---

> ### Author Response · Authors · 2023-11-29
> **RE: Reviewer 4Qb8**
>
> We thank reviewer 4Qb8 for support and their insight on issues of formalization and presentation.
>
> **Formalization**: We revised our definitions and cross-referenced our approach with multiple sources including Sontag, Kalman, and Ogata’s canonical control theory texts. Overall, we have shifted to a more reachability-first perspective — a point brought up in 4Qb8’s review. We have added a background section in Appendix A: “Abstract Systems an Control Theory Background” where we define generic systems, reachability, and controllability. We hope this will help communicate control theoretical concepts to newcomers properly. We also updated our formalization of LLM systems in Section 3. This update aims to align with the abstract systems and control definitions in the Appendix and to clarify the dynamical nature of LLM systems. We also added the following explanation in Section 3 about the key differences between LLM systems and more traditionally studied dynamical systems:
>
> “While LLMs are at times leveraged in a manner that masks the iterative aspects of generation, the reality is that token generation and externally imposed ``control input'' sequences are generated and processed sequentially, leading to non-trivial system dynamics.
>
> Several key differences remain between LLM-based systems and systems typically modeled through ordinary differential equations (ODEs), which have long been a cornerstone in the study of continuous-time dynamical systems:
> 1. **Discrete state and time**: LLM systems operate on sequences of discrete tokens over a discrete time set, in contrast to the continuous state spaces and time sets studied in classical control theory.
> 2. **Shift-and-Grow State Dynamics**: Whereas the system state in an ODE-based system has a fixed size over time, the system state x(t) for LLM systems grows as tokens are added to the state sequence.
> 3. **Mutual exclusion on control input token vs. generated token**: The LLM system state x(t) is written to one token at a time. The newest token is either drawn from the control input u(t) or is generated by the LLM by sampling x′ ∼ PLM (x′∣x(t)). This differs from traditional discrete stochastic systems, where the control sequence and internal dynamics generally affect the state synchronously.”
>
> __Input-output system and V^*__: While our empirical work revolves around “single-step” control (i.e., controlling the immediate next token generation), we have updated our LLM system definition and our characterization of LLM control theory to emphasize that LLM systems are, in fact, dynamical. Single-step control is merely a first step toward understanding the nature of more dynamical LLM system formulations with non-trivial internal dynamics (e.g., user interaction, chain-of-thought, retrieval-augmented generation, tool-wielding LLMs, etc). In particular, we acknowledge that the “shift-and- grow” state dynamics in V^* may be unfamiliar, but they are more realistic for how modern transformer-based LLMs function. This is particularly important for incorporating our analytic work on the reachable set for self-attention in Section 4. The growing state of LLMs is a fact of their operation, so to integrate our empirical work with our theoretical work, we maintain the state space V^*. Here is a paragraph from Section 6 discussing the relationship between the self-attention reachable set and multi-token generation:
>
> “Bounding the reachable set for self-attention is deeply related to the mechanism by which consistent representations are formed for multi-token generation.
>
> Steering a language model to generate a desired token sequence requires that the control input induce a token representation in the right-most token such that the next token prediction logits P (y∣u + x0 ) achieves a desired value.
>
> Moreover, generated tokens are fed back into the model, and their representations must be steered to control iterated generation.
>
> Self-attention is the primary mechanism by which the token representations exchange information, making the reachable set of output representations across multiple tokens in $\mathbf X_0$ for self-attention a fundamental part of LLM control theory.”
>
> Despite the growing state, we hope we have communicated why our LLM system formulation is still well-characterized as dynamical.
>
> We very much appreciate the feedback from reviewer 4Qb8 and we hope that our revised organization, structure, and expanded results address the concerns raised. As we are motivated by both the theoretical benefits of a control theoretic framework and the practical problems faced in LLM systems design, we hope that the reviewers enjoy the balance we seek to strike between the two.

---

### Meta-Review · Area_Chair_Hxzb · 2023-12-01

**Metareview:**

This paper uses concepts from control theory to explore the existence of special prompts that lead to controllable text generation. In particular, the authors introduce a notion of k-epsilon controllability which measures the degree to which such prompts exist. They explore this notion both theoretically and empirically, strongly suggesting that such prompts exist.

The idea of introducing concepts form control theory to large language model analysis is very welcome. However, the reviewers had serious concern with the formalization, the non-trivial conclusions that the paper allows to draw about controllability in practical settings (while the theoretical result seems so general that it might not be of interest to the ICLR community), and the novelty of the idea of using control theory in this setup, with respect to Soatto's et al "Taming AI Bots" paper (which is, however, to the best of my knowledge, only published on arXiv).

Overall, while I hope the authors will keep exploring the application of control theory to large language model analysis, it is not clear to me that the current report would be of broad interest to the ICLR community.

NB: The authors provided a rebuttal, but this was submitted much later than the rebuttal deadline, and it was thus not taken into account by the reviewers.

**Justification For Why Not Higher Score:**

While the idea of using concepts from control theory in LLM analysis is very welcome, this paper doesn't provide a convincing case for it.

**Justification For Why Not Lower Score:**

N/A

---

### Decision · Program_Chairs · 2024-01-16

Reject